# Personalization of Mechanical Ventilation Treatment using Deep Conservative Reinforcement Learning

## Abstract

Mechanical ventilation is a key form of support for patients with pulmonary impairment. Nonetheless, the optimal treatment regime is often unknown, leading to sub-optimal care and increased risks of complications. This work aims to develop a decision support tool to personalize mechanical ventilation. We present DeepVent, an off-policy deep reinforcement learning model that determines the best ventilator settings throughout a patient's stay. We evaluate our model using Fitted Q Evaluation, and show that it is predicted to outperform physicians. Moreover, we address the challenge of policy value overestimation in out-of-distribution settings using Conservative Q-Learning and show that it leads to safe recommendations for patients. We also design an intermediate reward based on the Apache II score to further improve our model's performance.

## 1. Introduction

The COVID-19 pandemic has put enormous pressure on the healthcare system worldwide, particularly on intensive care units (ICUs). In cases of severe pulmonary impairment, mechanical ventilation assists breathing in patients and acts as the key form of life support. However, the optimal mechanical ventilation regime is often unknown (Zein et al., 2016; Zampieri, 2017). This is due to the heterogeneity of medical histories and underlying conditions across patients (Patrick R Lawler, 2018). Furthermore, even if physicians knew the optimal regime for every patient, the overcrowding of ICUs often makes it impossible to attend to each patient in a timely manner. Sub-optimal mechanical ventilation treatment not only impedes recovery but can also lead to various complications including ventilator induced lung injury (VILI), ventilator-induced diaphragm dysfunction, pneumonia and oxygen toxicity (Pham et al., 2017). To prevent these complications, and offer optimal care, it is essential to personalize mechanical ventilation.

Reinforcement Learning (RL) is a subfield of machine learning used to solve sequential decision-making problems, which has gained popularity in healthcare in recent years (Raghu et al., 2017a; Prasad et al., 2017; Lin et al., 2018; Peng et al., 2019; Peine et al., 2021). By using observations of a patient's physiological states, the physician's actions and the corresponding outcomes, tools for treatment optimization can be developed.

In this paper we present DeepVent, a reinforcement learning model that we developed to optimize mechanical ventilation treatment. Our work makes three key innovations:

- We propose the first deep reinforcement learning approach to personalize mechanical ventilation settings

- We demonstrate the potential of Conservative Q-Learning, a recently proposed deep reinforcement learning algorithm, to address overestimation of the values of out-of-distribution states/actions, which is very important in a healthcare context, where data is limited and risk in decision making must be avoided

- We introduce an intermediate reward function based on the Apache II mortality prediction score to address the challenge of sparse reward in reinforcement learning

We compare DeepVent's decisions to those of physicians, as recorded in an existing standard dataset, as well as to those of an agent trained with Double Deep Q-Learning (DDQN), another popular deep reinforcement learning algorithm previously used in applications such as sepsis treatment (van Hasselt et al., 2015; Raghu et al., 2017b). According to our evaluation, DeepVent is predicted to outperform physicians, all while avoiding the overestimation problems of DDQN, thus making safe recommendations for patients.

### 1.1. Related Work

In recent years, important efforts have been made to implement machine learning as a decision support tool for mechanical ventilation. Here, we review relevant work in this domain.

**Ventilator settings optimization at individual timesteps.**
Kwok et al. developed a hybrid algorithm to predict ventila-

tion settings at a fixed timestep (Kwok et al., 2004). Akbulut et al. proposed a neural network model for a similar purpose (Akbulut et al., 2014). Recently, Venkata et al. applied inverse mapping of neural networks for ventilator settings prediction (Venkata et al., 2021). These works permit some amount of optimization. Nonetheless, they ignore the sequential nature of ventilation, which plays an important role in optimal recommendations (Peine et al., 2021).

**RL for optimization of ventilator settings.** Various works have used RL for mechanical weaning, the removal of patients from ventilators (Krinsley et al., 2012; Prasad et al., 2017; Yu et al., 2020). However, the optimization of mechanical ventilation settings using RL was only addressed last year when a tabular RL approach was used to predict the optimal levels for 3 settings (Peine et al., 2021). This offered a foundation for the use of RL in mechanical ventilation.

## 2. Preliminaries

### 2.1. Reinforcement Learning (RL)

The primary goal of RL is to train a policy $\pi$ that maximizes the return received from an environment. The environment is usually modeled by a *Markov Decision Process* (MDP), which is defined by a tuple $(\mathcal{S}, \mathcal{A}, P, r, \gamma)$.

- $\mathcal{S}$ : state space of the environment.

- $\mathcal{A}$ : action space of the environment

- $P$ : transition function, where $P(s_{t+1}|s_t, a_t)$ is the probability of arriving in state $s_{t+1}$ after taking action $a_t$ from state $s_t$.

- $r$ : reward function, where $r(s_t, a_t, s_{t+1})$ is the expected reward received by the policy $\pi$ after taking action $a_t$ from state $s_t$ and ending up in state $s_{t+1}$.

- $\gamma \in (0, 1)$ : discount factor of the reward.

At each time step $t$ of an episode, the policy observes the current state $s_t \in \mathcal{S}$, takes an action $a_t \in \mathcal{A}$, and ends up in another state $s_{t+1} \in \mathcal{S}$ while receiving a reward $r_t = r(s_t, a_t, s_{t+1})$. The goal of the policy is to maximize the cumulative discounted reward $R = \sum_{t=0}^{T} \gamma^t r_t$ received over the course of an episode with $T$ timesteps.

### 2.2. Q-Learning

A well-known RL algorithm is Q-learning (Watkins & Dayan, 1989), which aims to estimate the value of taking an action $a$ from a state $s$, known as the Q-value $Q(s, a)$.

At each timestep $t$, Q-learning takes some action $a_t$ from the state $s_t$ and arrives in a state $s_{t+1}$, where it receives a reward $r_t$, updating the Q value for $(s_t, a_t)$ as follows:

$$Q(s_t, a_t) = Q(s_t, a_t) + \eta(r_t + \gamma \max_a Q(s_{t+1}, a) - Q(s_t, a_t)) \tag{1}$$

where $\eta \in (0, 1)$ defines the learning rate. The intuition behind this is that we use the information received from the reward at the current timestep to update $Q(s_t, a_t)$ to be closer to a target value

$$r_t + \gamma \max_a Q(s_{t+1}, a) \tag{2}$$

which is a better estimate of the true value of $Q(s_t, a_t)$.

When the number of states is intractable, a deep Q-Network (DQN) algorithm is used (Mnih, 2015), wherein a neural network $Q_\theta$ (Q-network) with parameters $\theta$, and target network $Q_{\theta'}$ with parameters $\theta'$ are trained to output the value associated to any given state-action pair. The target network $Q_{\theta'}$ computes the target 2, which is used to update $Q_\theta(s_t, a_t)$ to be closer to that target. This is done by minimizing the mean squared Bellman error (MSBE) loss function

$$L_{DQN}(\theta) = \mathbb{E}_{s_t, a_t, r_t, s_{t+1} \sim D}[(r_t + \gamma \max_a Q_{\theta'}(s_{t+1}, a) - Q_\theta(s_t, a_t))^2] \tag{3}$$

The parameters $\theta'$ of the target network are periodically updated to be the same as $\theta$.

### 2.3. Double Deep Q-Networks (DDQN)

Overestimation occurs when the estimated value of a random variable is higher than its true value. DQNs were found to substantially overestimate the values of certain state-action pairs because, at timestep $t$, the value $Q(s_t, a_t)$ of a state-action pair is updated towards the target (2) which includes the maximum Q-value of the next state $s_{t+1}$ over all actions $a$.

DDQNs were introduced as a solution to this overestimation by modifying the calculation of the target 2. (van Hasselt et al., 2015). While DQNs use the target network to choose the maximum value action at the next state and to estimate the value of that action, DDQN uses two different networks, parametrized by $\theta$ and $\theta'$, one to choose the maximum value action at the next state and one to estimate the value of that action. At any point in time, one of the networks, chosen at random, is updated, by using as target the estimate from the other network. Thus, for network $Q_\theta$, the target (2) from DQN is replaced by $r_t + \gamma Q_{\theta'}(s_{t+1}, \text{argmax}_a Q_\theta(s_{t+1}, a))$ so that the final loss function to be minimized is

$$L_{DDQN}(\theta) = \mathbb{E}_{s_t, a_t, r_t, s_{t+1} \sim D}[(r_t + \gamma Q_{\theta'}(s_{t+1}, \text{argmax}_a Q_\theta(s_{t+1}, a)) - Q_\theta(s_t, a_t))^2] \tag{4}$$

Although this partially solves the overestimation problem of DQNs, DDQNs can still suffer from some overestimation and thus only partly overcome this challenge (van Hasselt et al., 2015).

## 2.4. Offline Reinforcement Learning

To further address the overestimation challenge, we have to first understand offline RL. Traditional RL methods are based on a fundamentally online learning paradigm, by which an agent actively interacts with an environment, receives a reward, and updates its policy accordingly. This is an important barrier to RL implementation in many fields, including healthcare (Levine et al., 2020), where acting in an environment is not only inefficient, but also unethical, as it puts patients at risk. Consequently, recent years have witnessed important growth in offline (also known as batch) RL, where the learning is driven from a dataset of transitions $\mathcal{D} = \left\{ \left( s_t^i, a_t^i, r_t^i, s_{t+1}^i \right) \right\}_{i=1}^N$.

When applied to an offline setting, RL methods often exhibit poor performance because their understanding of the environment is limited to the used dataset. This can lead them to overestimate the Q-values of state-action pairs which are underrepresented in the dataset, or out-of-distribution (OOD). Since the policy derived from these methods chooses the action with the highest Q value at each state, this can lead to sub-optimal action choices (Kumar et al., 2020). In the healthcare setting, this can translate to unsafe recommendations, putting patients at risk.

## 2.5. Conservative Q-Learning (CQL)

To address the challenge of overestimation in an offline setting, Conservative Q-Learning (CQL) was proposed with the objective of learning a conservative estimate of the Q-function (Kumar et al., 2020). This is done by adding a regularization term to the loss function of the Q-networks:

$$\mathbb{E}_{\mathbf{s_t} \sim \mathcal{D}, \mathbf{a_t} \sim A}[Q(\mathbf{s_t}, \mathbf{a_t})] \qquad (5)$$

which intuitively prevents Q-values from getting too high.

However, to prevent too much underestimation due to this added term, a maximization term is also added to the loss function:

$$- \mathbb{E}_{\mathbf{s_t}, \mathbf{a_t} \sim \mathcal{D}}[Q(\mathbf{s_t}, \mathbf{a_t})] \qquad (6)$$

In Eq.(6), the $Q$-values contributing to the expectation are only the ones for state-action pairs which are actually in the dataset, whereas in 5 the $Q$-values are for states in the dataset combined with **any** $a \in A$. CQL thus simultaneously minimizes the Q-values of all the actions in our action space while maximizing the Q-values of the actions which are most observed in the data. This effectively prevents overestimation of OOD actions and states which are underrepresented in the dataset.

CQL can be built on top of any deep RL method by simply adding these conservative terms (and a hyperparameter $\alpha$ to scale them) to its loss function.

## 3. Datasets

### 3.1. Patient Cohort and Data Collection

We used the MIMIC-III database (Johnson et al., 2016), an open-access database containing data regarding 61,532 ICU stays admitted to the Beth Israel Deaconess Medical Center (Boston, MA, USA) between 2001 and 2012. More details descriptions of the data can be found in Appendix A.

Standardized Query Language (SQL) was used to extract patient data from the MIMIC-III database into a table of four-hour time windows. For each patient, the following data were extracted: vital signs, lab values, demographics, fluids and ventilation settings (see 4.1 for detail). This resulted in a total of 19,780 ventilation events.

### 3.2. Preprocessing and imputation

Ventilation events were separated using their unique *icustay_id*. For each of them, the first 72 hours of ventilation were selected. The patient data was separated into parallel state, action and reward arrays. A parallel array filled with 0s and a 1 at the terminal state was instantiated to keep track of the trajectory's length.

For data imputation, a mix of methods were used. In the case where less than 30% of the data was missing, KNN imputation was used with $k = 3$. In the case where 30% to 95% of the data was missing, a time-windowed sample-and-hold method was used, by which we took the initial value and replaced the following values with it until either a new value was met or a limit was reached. When the initial value was missing, mean value imputation was performed. Finally, if over 95% of the data was missing, the variable was removed from our state space (Bertsimas et al., 2020).

### 3.3. Generation of the Out-of-distribution dataset

To investigate the overestimation of DeepVent and DDQN, an out-of-distribution (OOD) set of outlier patients was created. An outlier patient was defined as having at least one state feature (demographic, vital sign, lab value or fluid value) at the beginning of ventilation in the top or bottom 1% of the dataset distribution. Approximately 25% of patients were considered outliers.

## 4. Proposed Approach

### 4.1. RL Problem Definition

Similar to (Peine et al., 2021), we defined an episodic problem with a finite horizon, where each episode lasts from the time of the patient's intubation to 72 hours after.

**State Space** The state space $\mathcal{S}$ was built from the following variables:

- Demographics: Age, gender, weight, readmission to the ICU, Elixhauser score

- Vital Signs: SOFA score, SIRS score, GCS score, heart rate, systolic BP, diastolic BP, mean BP, shock index, respiratory rate, temperature, spO2

- Lab Values: Potassium, sodium, chloride, glucose, bun, creatinine, magnesium, carbon dioxide, hemoglobin, white blood cell count, platelet count, partial thromboplastin time, prothrombin time, international normalized ratio, pH, partial pressure of carbon dioxide, base excess, bicarbonate

- Fluids: Urine output, vasopressors, intravenous fluids, cumulative fluid balance

**Action Space** The 3 ventilator settings of interest are:

- Adjusted tidal volume *or* $Vt$ (Volume of air in and out of the lungs with each breath adjusted by ideal weight)
- PEEP (Positive End Expiratory Pressure)
- FiO2 (Fraction of inspired oxygen)

The action space $\mathcal{A}$ is the Cartesian product of the set of these three settings. Each setting can take one of seven values corresponding to ranges. We can therefore represent an action as the tuple $a = (v, o, p)$ with $v \in Vt, o \in FiO_2, p \in PEEP$.

Here are the corresponding ranges:

Vt (ml/Kg): [0-2.5, 2.5-5, 5-7.5, 7.5-10, 10-12.5, 12.5-15, >15]
PEEP (cmH2O): [0-5, 5-7, 7-9, 9-11, 11-13, 13-15, >15]
FiO2 (%): [25-30, 30-35, 35-40, 40-45, 45-50, 50-55, >55]

**Reward Function** The main objective of our agent is to keep a patient alive in the long-term. Therefore, even if DeepVent only treats patients for 72 hours, it learns how to maximize their 90 day survival. This way, DeepVent aims to comprehend not only the immediate consequences of its actions, but also the long term complications that could arise. It then learns how to prevent these complications and maximize long-term survival. An obvious reward function $r$ for this problem is a terminal reward $r_1(s_t, a, s_{t+1})$, which takes the value $-1$ if $s_{t+1}$ is the final state in an episode corresponding to a patient who died, and $+1$ if $s_{t+1}$ is the final state in an episode corresponding to a patient who was still alive or discharged 90 days after admission to the ICU.

The sole use of a sparse terminal reward is known to cause poor performance in RL tasks (Mataric, 1994). We therefore developed an intermediate reward based on the Apache II score, a widely used score in ICUs to assess the severity of a patient's disease (Knaus et al., 1985). This score was slightly modified to be best adapted to our dataset.

Our final modified Apache II score was based on temperature, mean blood pressure, heart rate, pH, sodium level, potassium level, creatinine level and white blood cell count. The contributions of these parameters to the final score were the same as for the original Apache II score.

In order to not simply define reward based on how well a patient was doing but rather their evolution through time, our intermediate reward consists of the negative difference in the modified Apache II score between states $s_t$ and $s_{t+1}$, which is normalized by dividing it by the total range of the score. Combining our intermediate reward with the terminal reward, we obtain our final reward function:

$$r(s_t^i, a_t^i, s_{t+1}^i) = \begin{cases} +1 & \text{if } t+1 = l_i \text{ and } m_{t+1}^i = 1 \\ -1 & \text{if } t+1 = l_i \text{ and } m_{t+1}^i = 0 \\ \frac{(A_{t+1}^i - A_t^i)}{\max_A - \min_A} & \text{otherwise} \end{cases}$$

where:

- $A_t^i$ is the modified Apache II score of patient $i$ at timestep $t$

- $m_t^i = 0$ if patient $i$ is dead at timestep t and 1 otherwise

- $l_i$ is the length of patient $i$'s stay at the ICU

- $\max_A, \min_A$ are respectively the maximum and minimum possible values of our modified Apache II score

**Transition Function** The transition function $P$ is not known due to the model-free nature of our approach.

### 4.2. RL Algorithm Implementation

Our implementation of CQL was built on top of a DDQN implementation to facilitate comparison between the two algorithms. Both come from the offline Deep Reinforcement Learning Library D3RLPY (Takuma Seno, 2021).

Since our CQL implementation was built on top of DDQN, the loss function it minimized was

$$L_{CQL}(\theta) = \alpha(\mathbb{E}_{\mathbf{s} \sim \mathcal{D}, \mathbf{a} \sim A}[Q_\theta(\mathbf{s}, \mathbf{a})] - \frac{1}{2}\mathbb{E}_{\mathbf{s_t}, \mathbf{a_t} \sim \mathcal{D}}[Q_\theta(\mathbf{s_t}, \mathbf{a_t})]) + L_{DDQN}(\theta) \quad (7)$$

where $L_{DDQN}(\theta)$ is defined in Eq. (4) and $\alpha$ is the hyperparameter which scales the weight of the conservative term in the loss.

### 4.3. Off-Policy Evaluation

In online RL, policies are typically evaluated through interaction with the environment. However, in the healthcare setting where the environment is real patients, evaluating the policies in this manner would be unsafe. Evaluation is therefore done by using the dataset through various methods grouped under the term Off-Policy Evaluation (OPE).

The performance of these various methods was recently evaluated in the healthcare setting (Tang & Wiens, 2021), leading to the conclusion that Fitted Q Evaluation (FQE) consistently provided the most accurate results. Following this reasoning, we use FQE from D3RLPY (Takuma Seno, 2021) to evaluate our policies. FQE takes as input a dataset $D$ and a policy $\pi$, and outputs a value estimate for each state in $D$. This estimate corresponds to an approximation of the cumulative discounted reward which would be received by a policy $\pi$ when applied to the given state (Le et al., 2019).

## 5. Results & Discussion

The following results are all averaged over 5 independent runs with different train/test splits using the best hyper-parameters found after a thorough hyper-parameter search. Variances are included when appropriate. More details about the training of our models can be found in Appendix B.

### 5.1. DeepVent Overall Performance

To begin, we compare the performance of DeepVent- (CQL without intermediate reward), DeepVent (CQL with intermediate reward), and the physician when applied to the patients in our test set (see Table 1).

*Table 1.* Average initial state value estimates for physician, DeepVent- and DeepVent, with standard errors. DeepVent- significantly outperforms the physician. The addition of the Apache II derived intermediate reward leading to DeepVent further improves the estimate.

| PHYSICIAN | DEEPVENT- | DEEPVENT |
|---|---|---|
| $0.502 \pm 0.00709$ | $0.762 \pm 0.00402$ | $0.797 \pm 0.00670$ |

The initial state of an episode in our test set represents the state of a given patient when ventilation is initiated. The performance of DeepVent- or DeepVent when faced with treating that patient can be approximated by the value estimation output by FQE for the initial state of that patient.

Although DeepVent was trained with intermediate rewards, the value estimation by FQE only depends on the dataset $\mathcal{D}$ and the actions chosen by the policy $\pi$ used to train FQE. Since $\mathcal{D}$ has no intermediate reward, the estimates are solely based on the terminal reward and can thus be used as a fair comparison between DeepVent- and DeepVent.

Since the physician policy effectively generates the episodes in our dataset, its value estimates for each initial state can be computed by taking the cumulative discounted reward for the episode in our dataset starting at that initial state.

Using these estimates, we observe that DeepVent outperforms physicians by a factor of 1.52. The addition of the intermediate reward increases this factor to 1.59.

### 5.2. DeepVent and Safe Recommendations

To evaluate DeepVent's recommendations, we here compare them to DDQN. We first evaluate their respective similarity to physicians in terms of settings chosen (see Figure 1).

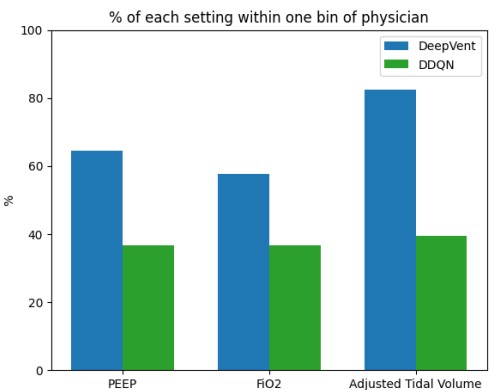

*Figure 1.* % of states for which the algorithm's recommendation is within one bin of the physician's recommendation (see binning process in 4.1). As compared to DDQN, DeepVent suggests actions more similar to the physicians.

In comparison to DDQN, DeepVent was found to choose actions closer to the physicians for all three parameters. This suggests that DeepVent learns actions that are more clinically relevant and potentially safer for patients. In order to confirm this hypothesis and further understand the nature of these differences, the distribution of these actions across the different policies were investigated (see Figure 2).

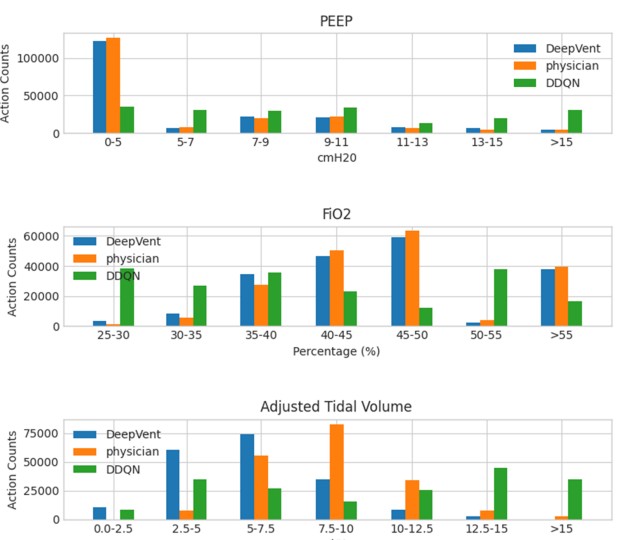

*Figure 2.* Distribution of actions across ventilator settings. Unlike DDQN, DeepVent makes recommendations in safe and clinically relevant ranges for each setting.

DeepVent was observed to suggest safer setting recommendations when compared to DDQN. The standard of care in

terms of PEEP setting is commonly initiated at 5 cmH2O (Nieman et al., 2017) which is supported by the high number of recommendations by physicians being in the range of 0-5 cmH2O in our dataset. DeepVent spontaneously chose to adopt this strategy by making most recommendations in the range of 0-5 cmH2O. In contrast, DDQN chose settings distributed along all the options, ranging up to 15 cmH2O, where physicians in our dataset rarely went. While the optimal setting for PEEP is still a topic of debate (Nieman et al., 2017), high PEEP settings have been associated with higher incidence of various complications, including pneumothorax (Zhou et al., 2021), inflammation (Güldner et al., 2016) and impaired hemodynamics (PROVE Network Investigators for the Clinical Trial Network of the European Society of Anaesthesiology et al., 2014), and should therefore be avoided.

In terms of FiO2 setting, DeepVent was once again found to follow clinical standards of care. More specifically, we observe that DeepVent often chose actions in the same ranges as physicians in our dataset, with many recommendations in the ranges of 35-50% and >55% and few recommendations below 35% and between 50-55%. In contrast, DDQN made few recommendations in ranges often suggested by physicians, and many in those that were rarely employed.

Finally, we found that DeepVent suggested better recommendations for the adjusted tidal volume when compared to DDQN. One of the most common respiratory conditions requiring ventilation is known as acute respiratory distress syndrome (ARDS). Various studies have shown that in patients with ARDS, the optimal tidal volume is found in the range of 4-6 ml/kg (Retamal et al., 2013; Luks, 2013). In patients without ARDS, clinical trials have shown that the optimal range is 6-8 ml/kg (Jaswal et al., 2014; Kilickaya & Gajic, 2013). Overall, the standard of care in terms of adjusted tidal volume is around 6 ml/kg. DeepVent made a majority of recommendations in the range of 2.5-7.5 ml/kg, with an important amount of these being concentrated in the 5-7.5 ml/kg range. In contrast, DDQN made many recommendations in higher ranges, going as far as equal to or above 15 ml/kg, values rarely observed in clinical practice and associated with increased risks of lung injury and mortality (Serpa Neto et al., 2012).

We therefore conclude that in contrast to DDQN, DeepVent chooses actions in clinically relevant and safe ranges. The superior estimated performance of DeepVent when compared to physicians (section 5.1) likely comes from the fact that it is capable of constantly monitoring a large scale of data, from demographics and vital signs to lab values and fluids. In addition, by being a deep RL model, DeepVent learns the intricate consequences of its actions and may thus be able to predict complications before they even arise.

We then investigated the correlations between differences

in actions between the RL algorithms and the physician for each ventilation setting (Vt, $FiO2$, and PEEP) and observed mortality (see Figure 3). Specifically, we computed the difference in bins, between the actions selected by the policy of the given algorithm (either CQL or DDQN) and the actions of the physician, across all trajectories. Similar u-curves can be found in (Raghu et al., 2017a) and (Gottesman et al., 2018). We note that u-curves are not fully representative of a policy's performance and should be investigated with caution (Gottesman et al., 2018). Nonetheless, they can contribute interesting insight into the correlations between action choices and survival.

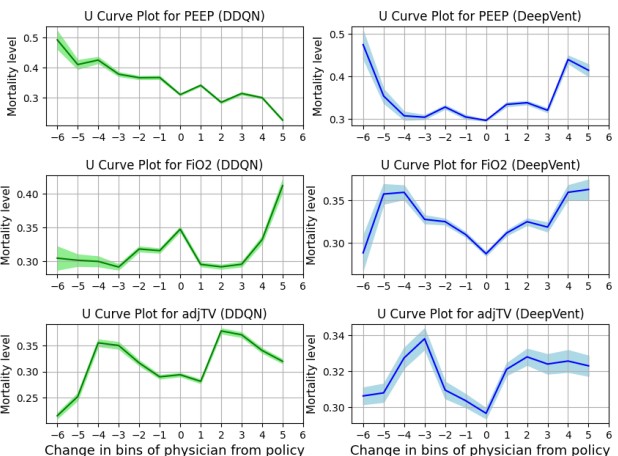

Figure 3. U-curves for DDQN and DeepVent policies. Observed mortality is plotted against the difference in actions between the RL algorithm and the physician (calculated as the bin of RL agent's action − bin of physician's action). Lowest mortality for DDQN is achieved when it takes actions highly different from the physician's recommendations. In contrast, lowest mortality for DeepVent is observed when it picks actions in the same bin as physicians, strengthening the conclusion that DeepVent choosing clinically-relevant actions leads to higher survival

We observe that whenever physicians and DeepVent act similarly, observed patient mortality tends to be at its lowest point across each action setting. In addition, mortality generally increases as the physician's actions get further from DeepVent's. This suggests that DeepVent's policy choosing actions similar to physicians leads to optimal survival. In contrast, when physicians and DDQN act similarly, we observe sub-optimal mortality. DDQN has to choose actions very different from the physicians' to achieve a similar expected survival, which is suspicious since that would go against the many clinical trials discussed previously.

## 5.3. DeepVent in Out-Of-Distribution Samples

We next investigated whether the sub-optimal recommendations made by DDQN may be caused by value overesti-

mation. To do so, we investigated the mean initial Q values for DeepVent and DDQN (as estimated by FQE). It is interesting to not only understand how well the model performs on data similar to that on which it was trained, but also on outlier data. In healthcare in particular, a model may face patients different than those on which it was trained, and assuring it is still reliable in this setting is essential for safe implementation. We thus consider both an in-distribution (ID) and an out-of-distribution (OOD) setting (see Figure 4).

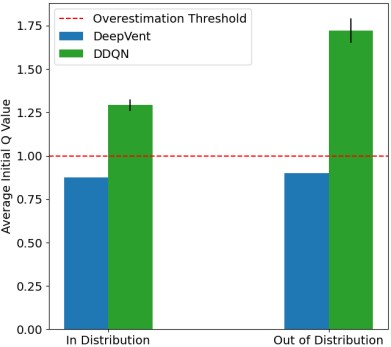

*Figure 4.* Mean initial Q-values for both in and out of distribution settings for DeepVent and DDQN (with variances - DeepVent's variance is not visible because it is so small). The horizontal line is the maximum expected return per episode. In contrast to DeepVent, DDQN clearly suffers from overestimation, which is aggravated in the OOD setting

Since the maximal expected return for an episode in our dataset is set at 1, any value for DDQN above this threshold should be considered as overestimated. We observe that DDQN overestimates policy values in both the ID and OOD settings. In addition, we observe that DDQN assigns higher values to its initial states in the OOD setting, suggesting it believes that performance is better when facing outliers than patients similar to the ones on which it was trained. This lack of capacity to accurately assess its initial state values may be the cause of its unsafe recommendations discussed above.

Meanwhile, DeepVent seems to avoid these problems, as its average initial state value estimate stays below the maximal overestimation threshold of 1 in both settings, and barely increases in the OOD setting when compared with the ID setting.

## 6. Conclusion

In this work, we developed DeepVent, a decision support tool for personalizing mechanical ventilation treatment using deep reinforcement learning. We showed that our use of Conservative Q-Learning leads to settings in clinically relevant and safe ranges by addressing the problem of overestimation of the values of out-of-distribution state-action pairs. Furthermore, we showed using FQE that DeepVent achieves a higher estimated performance when compared to physicians, which can be further improved through the implementation of an intermediate reward based on the Apache II mortality prediction score. We conclude that DeepVent intuitively learns to pick actions that a physician would agree with, while using its capacity to overview vast amounts of clinical data at once and understand the long-term consequences of its actions to improve outcomes for patients. Moreover, the fact that DeepVent is associated with low overestimation in out-of-distribution settings makes it much more reliable, and thus closes the gap between research and real-world implementation. Future work should aim to investigate the potential of the DeepVent methodology in other healthcare applications.

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

# Appendix

## A. Dataset

We used the MIMIC-III database (Johnson et al., 2016), a free and publicly-available database of deidentified health data for over 40,000 patients at the Beth Israel Deacones Medical Center from 2001 to 2012. Below, we include the table of features extracted from MIMIC-III. Big Query and SQL were used to for the extraction into a comma separated value (CSV) file.

*Table S1.* Description of the features of the used dataset

|  | category | description |
| --- | --- | --- |
| icustay_id | - | ID of each icu admission |
| subject_id | - | ID of each patient |
| hadm_id | - | ID of each hospital admission |
| start_time | - | start time of a 4 hour bloc |
| first_admit_age | demographics | age |
| gender | demographics | - |
| weight | demographics | - |
| icu_readm | demographics | Readmission to ICU (T/F) |
| elixhauser_score | demographics | Elixhauser score |
| sofa | vital signs | SOFA score |
| sirs | vital signs | SIRS score |
| gcs | vital signs | GCS score |
| heartrate | vital signs | heart rate |
| sysbp | vital signs | systolic BP (blood pressure) |
| diasbp | vital signs | diastolic BP (blood pressure) |
| meanbp | vital signs | mean BP (blood pressure) |
| shockindex | vital signs | shock index |
| resprate | vital signs | respiratory rate |
| tempc | vital signs | temperature (*C) |
| spo2 | vital signs | spo2 |
| potassium | lab values | - |
| sodium | lab values | - |
| chloride | lab values | - |
| glucose | lab values | - |
| bun | lab values | blood urea nitrogen |
| creatinine | lab values | - |
| magnesium | lab values | - |
| carbondioxide | lab values | - |
| hemoglobin | lab values | - |
| wbc | lab values | white blood cell count |
| platelet | lab values | - |
| ptt | lab values | partial thromboplastin time |
| pt | lab values | prothrombin time |
| inr | lab values | international normalized ratio |
| ph | lab values | pH |
| paco2 | lab values | partial pressure of carbon dioxide |
| base_excess | lab values | - |
| bicarbonate | lab values | - |
| mechvent | settings | mechanical ventilator (on/off) |
| fio2 | settings | FiO2 level |
| urineoutput | fluids | - |
| vaso_total | fluids | total amount vasopressors |
| iv_total | fluids | total amount intravenous |
| cum_fluid_balance | fluids | cumulative fluid balance |
| peep | settings | positive end-expiratory pressure |
| tidal_volume | settings | - |
| hospmort90day | outcome | hospital mortality within 90 days |
| dischtime | - | discharge time |
| deathtime | outcome | death time |

## B. Training Details

We first split our preprocessed episodes into a training (80%) and validation (20%) set. We then conducted an initial grid search to find the best hyper-parameters for our model. The main hyper-parameters were the learning rate $\eta$, the discount factor $\gamma$, and, for CQL, the scaling factor $\alpha$ for the conservative part of the loss function. For both CQL and DDQN, we used Q-networks with 1 hidden layer of 256 neurons.

We considered the $\eta$ values $[1^{-7}, 1^{-6}, 1^{-5}, 1^{-4}]$ the $\gamma$ values $[0.25, 0.5, 0.75, 0.9, 0.99]$, and the $\alpha$ values $[0.05, 0.1, 0.5, 1, 2]$. We then started by partially training a model using each combination of hyper-parameters for 500000 steps and observing the preliminary results.

Using this method, we determined that the best hyper-parameters were $\gamma = 0.75$ and $\eta = 1^{-6}$ for DDQN, and $\gamma = 0.75$, $\eta = 1^{-6}$ and $\alpha = 0.1$ for CQL. We then trained DeepVent-, DeepVent and DDQN on the training set for 5 runs of 2000000 steps each using these hyper-parameters and averaged the results for all our graphs. Each run was done with a different train-test split and took around 14 hours to complete on GeForce GTX Titan X GPUs.