# OpenReview forum: "Personalization of Mechanical Ventilation Treatment using Deep Conservative Reinforcement Learning"
_uoft.ai/University_of_Toronto/2021/ProjectX — ProjectX2021_

### Official Review · Reviewer_rLge · 2022-02-09
**Reinforcement learning strategies help determine optimal patient ventilation parameters.**

**Rating:** 7
**Confidence:** 3

**Review:**

- The project establishes a deep reinforcement learning approach to personalize mechanical ventilation settings in the clinic.
- The question has clinical significance, in the context of assisted diagnostic options presented to physicians.
- The context is clearly explained and motivated by a current review of clinical and technical literature.
- The RL approach seems state of the art, properly justified and technically sound.
- It was unclear to the reviewer how the datasets were split between training, validation and testing subsamples.
- The clarity of the presentation of the results could be improved, especially for the greater appreciation of a non technical reader.
- The results and limitations are properly discussed.

---

### Official Review · Reviewer_jH7e · 2022-02-10
**Ventilation treatment review**

**Rating:** 9
**Confidence:** 4

**Review:**

Connection to current science (2.5/3):
- Clear justification of need for this type of model
- Novelty of this approach is well outlined alongside limitations of previous similar work
- How would you see this tool being implemented in a healthcare setting?

Communication (1.75/2):
- Paper is quite dense but would be appropriate for a machine learning-focused journal. Using a bit less jargon would improve the potential reach of this work.
- Figures are clear and support the conclusions

Methodology (4/4):
- Methods are thorough and seem appropriate, the assumptions seem reasonable.
- Decision to consider both 72h treatment window + 90 day survival is wise
- Good use of physician performance as a benchmark and comparison to another model already in use


Reproducibility (0.75/1)

---

### Official Review · Reviewer_DtPn · 2022-02-11
**An innovative approach to existing systems for an incredibly relevant clinical dilemma**

**Rating:** 9
**Confidence:** 4

**Review:**

Connection to current science (2.5/3)
 - Clearly well research medical context as well as appreciation of how this will aid the strain on the medical system due to the pandemic
 - Outlines how this model would be applied in conjunction with physician guidance
 - Appreciate the concern for safety, long-term survival and ethics outlined in their explanation of how they were creating this model with a health and critical care focus in mind
 - Minor: Could have expanded more on patient outcomes at 90 days other than simply discharge or death, ventilator related lung injury is a major complication of COVID treatment and is a large factor guiding ventilation settings to optimise patient function even if discharged alive from the ICU

Clarity of communication (1.8/2)
 - Paper is very well written, figure captions are descriptive and detailed
 - Separation of results and discussion sections would be useful to outline objective findings of their research compared to more subjective comparisons to existing models and future directions

Methodological quality (3.5/4)
 - Excellent use of existing models and enhancing upon their flaws in a relevant clinical setting
 - Appropriately complex and takes into account a wide variety of patient data
 - Covers the topic of how this model will react to an 'unusual' patient
 - Needed to explain more about modified Apache II system and why certain metrics were removed and how this may affect their predictions

Reproducibility (1/1)
 - Open source health records
 - Used existing Q-learning algorithms and included appendix 2 to outline training of the model

---

### Official Review · Reviewer_jVnF · 2022-02-14
**Review—Personalization of Mechanical Ventilation Treatment using Deep Conservative Reinforcement Learning**

**Rating:** 8
**Confidence:** 3

**Review:**

The authors developed DeepVent, a reinforcement learning model to optimize ventilation treatment for patients with pulmonary impairment. The model was evaluated using Fitted Q Evaluation and compared against physicians actions’ and a trained Double Deep Q-Learning algorithm. Conservative Q-Learning was used to address overestimation in out-of-distribution settings. Overall, the authors conducted a methodologically ambitious paper and provided detailed explanations of their methods.

**Connection to Current Science (2.25/3)**

This study uses Reinforcement Learning to develop personalized mechanical ventilation recommendations. It builds on previous research by incorporating the sequential nature of ventilation to improve optimal recommendations. Although the paper provides ample technical descriptions, the authors could also discuss potential pathways to implementation to strengthen their study.

**Clarity of Communication (1.75/2)**

The authors describe their study well and visualize results using well-designed and labelled graphs. The methods are well explained and easy to follow. The clarity of the paper could be improved by providing more background information about mechanical ventilators and their clinical setting. For example, it’s unclear what constitutes a “ventilator event.” Additionally, it’s not clear at the start of the paper what ventilator settings are being optimized. In lines 68-71, “RL approach was used to predict the optimal levels for 3 settings,” it’s unclear what these three settings are. If these settings are PEEP, FiO2, and Adjusted Tidal Volume, I suggest introducing them earlier in the paper and providing this contextual information. In a health context, a descriptive Table 1 would be expected to show data distributions and provide insight into the sample’s demographic characteristics, which is especially useful when determining external validity. (1.75/2)

**Methodological Quality (3.5/4)**

The methodology was ambitious and described in detail. Two critical issues to consider are the 30-95% missingness and the use of lab values as predictive features.

1. A large amount of missing data should address data quality issues and whether missingness is informative or random. Additionally, more evidence is needed to support whether a “time-windowed sample-and-hold” method is appropriate for all features (i.e., what is the typical temporal variability of vital signs, lab values, and fluids?).

2. Consider a sensitivity analysis to determine the impact of including vs. excluding lab results into the model. The model’s performance over physicians may be over-optimistic because lab results (the actual processes and mechanisms associated with a lab requisition) may signal patient status and prognosis. Those who are receiving medical treatment (e.g., a lab requisition) are likely to be at higher risk of a clinical event, and the presence of laboratory test orders and their timing may actually be better predictors than the test results themselves (i.e., the lab result values). (See Agniel D, Kohane I S, Weber G M. Biases in electronic health record data due to processes within the healthcare system: retrospective observational study BMJ 2018; 361 :k1479 doi:10.1136/bmj.k1479)

Additionally, since the authors mention that ventilation regimes vary across heterogenous medical histories and underlying patient conditions and characteristics, it would be essential to assess how model performance generalizes across different patient populations and various medical conditions.

**Reproducibility (1/1)**

Authors provided supplementary material with well-documented code, enabling others to reproduce the paper’s results.

---

### Decision · Program_Chairs · 2022-02-19

Winner